# Assessment of Nutritional Quality of Products Sold in University Vending Machines According to the Front-of-Pack (FoP) Guide

**DOI:** 10.3390/nu14235010

**Published:** 2022-11-25

**Authors:** Carmen Lasala, Antonio Durán, Daniel Lledó, Jose M. Soriano

**Affiliations:** 1Area of Nutrition, University Clinic of Nutrition, Physical Activity and Physiotherapy (CUNAFF), Lluís Alcanyís Foundation-University of Valencia, 46020 Valencia, Spain; 2Food & Health Lab, Institute of Materials Science, University of Valencia, 46980 Paterna, Spain; 3Joint Research Unit on Endocrinology, Nutrition and Clinical Dietetics, Research Institute La Fe, University of Valencia-Health, 46026 Valencia, Spain

**Keywords:** vending machines, nutritional quality, food environment, consumer health, obesogenic environment

## Abstract

The aim of this research was to assess the nutritional composition of 654 foods and beverages from vending machines on a University Campus. The guide called “Creating a front of pack nutrition label for pre-packed products sold through retail outlets” from the Department of Health (UK) was used to assess the nutritional composition and to compare values from food and beverage categories. A high proportion of food items had high–moderate content of nutrients related to chronic disease (93, 88, 74 and 49% had high–moderate content of fat, saturated fat, salt and sugar, respectively). On the other hand, a few beverages had high–moderate content of these nutrients, except for sugar (2% high to moderate in fat, 2% in saturated fat, 0% in salt and 39% in sugar). The most frequent food categories were chocolates and bars (10.6%) and breadsticks (8.4%), whereas the most available beverages were water (27.7%) and soft drinks (23.4%). There were no statistically significant differences in the proportion of categories between the health science and humanities faculties, except for energy drinks (*p* < 0.05) and soft drinks (*p* < 0.05). Vending machines contribute to an obesogenic environment and do not support healthy snacking. Recommendations in this article should be considered to develop official guidelines to ensure the wider availability of healthy dietary choices in vending machines in university settings.

## 1. Introduction

Excess weight is a serious public health issue with significant associated mortality. Factors that have favored a positive balance of energy and weight gain in recent decades include the progressive increase in per capita food consumption, particularly of high-calorie and ultra-processed foods; the decrease in time devoted to physical activity; and the increase in sedentary activities such as watching television or using electronic devices [1]. Obesity is considered a major risk factor for numerous chronic health problems, including cardiovascular disease, diabetes mellitus, hypertension and some types of cancer [2]. Several long-term epidemiological studies have shown that obesity is strongly associated with an increased risk of mortality [1]. In 2018, 17.4% of the Spanish population over 18 years old and 10.3% of the child population had obesity, increasing in frequency as social class decreases. If we consider obesity and overweight together, more than half of adults (54.5%) and more than a quarter of the child population (28.6%) suffer from being overweight [3]. The latest guidelines on cardiovascular prevention emphasize the promotion of healthy lifestyles, including the importance of an adequate diet and limiting the intake of salt, sugars and saturated fats, among other measures [2]. The acquisition of these healthy habits is even more important in young people since inadequate dietary patterns at an early age will have a negative impact on their health in the future [4]. 

Some studies have emphasized the potential role of vending machines (VMs) in eating behaviors and dietary quality as they facilitate the availability of food and beverages in schools, universities, hospitals and work settings [4,5]. Vending machine behaviors, such as vending machine accessibility and use, have been positively associated with the consumption of snacks and soft drinks [6]. Some studies have analyzed the nutritional quality of vending machine products, showing a high percentage of energy as well as fats, saturated fats, sugars and salt [4,5,6,7,8], which favors the development of overweight and obesity and, consequently, enhances the risk of suffering from chronic health problems [2]. It has been observed that the use of vending machines is higher among younger people. In 2018, in Spain, 6 out of 10 young people aged 18 to 24 reported having used vending in the last 6 months [9]. Thus, analyzing the products available in VMs provides an accurate picture of students’ intake. Consequently, improving the nutritional quality of VMs in Universities would constitute a preventive measure against cardiovascular and metabolic diseases. 

Given the limited evidence regarding the nutrient content of foods sold in vending machines in Spanish Universities and their potential role in university students’ food intake and body weight, the primary aim of the current study was to assess the nutritional value of products sold in vending machines in a Spanish University campus, to establish recommendations to improve the nutritional profile of the products offered, thus increasing healthy choices.

## 2. Materials and Methods

### 2.1. Sampling and Variables

Vending machines (VMs) containing food and beverages located on a University Campus were analyzed. Two trained researchers identified fourteen VMs located on the Campus, nine from health sciences faculties (Faculty of Medicine, Psychology and Sports Sciences) and five from humanities (Faculty of History, Philology and Philosophy). Empty spaces in the VMs were not taken into account, valuing only those slots with available products. Photographs were taken to identify the products by name and weight. The information from nutrition labels and the ingredient listing from a total of 654 products identified with 120 unique product labels was compiled in an excel file by two independent researchers and included 85 types of foodstuffs classified into 11 categories (biscuits, breadsticks, candies and sweets, chewing gums, chocolates and bars, fruit, nuts, pastries, potato chips, sandwich and snacks) and 35 types of beverages in 5 categories (energy drinks, juices, milkshakes, soft drinks and water). The nutritional quality was evaluated by assessing several variables from the excel file. The study variables are food and beverage categories, nutrient content (energy, fat, saturated fat, carbohydrates, sugars, fiber, protein and salt) and proportions of products with palm oil and gluten. The nutrient content is expressed per 100 g or 100 mL and per package/bottle weight. We considered the whole package as the serving size because the purchase of these products usually entails its entire consumption on the same day.

### 2.2. Grouping of Food and Beverage Items

Food items were grouped into eleven categories (nuts; fruit; chocolates and bars; pastries; breadsticks, potato chips; snacks; candies and sweets; chewing gum; sandwiches; and biscuits) and beverages into five categories (water; soft drinks; milkshakes; juices and; energy drinks), according to the nature of the product. 

### 2.3. Nutritional Profiling

Nutrient content was evaluated according to the Guide “Creating a front of pack (FoP) nutrition label for pre-packed products sold through retail outlets” [10]. Thus, products were classified as high, medium or low, considering fat, saturated fat, sugar and salt content per 100 g or 100 mL (Table 1). 

### 2.4. Statistical Analysis

Variables were analyzed using descriptive statistics indicators (absolute frequencies, percentages, means and standard deviations). Pearson’s Chi-square test (X2) was used to evaluate the differences between qualitative variables. SPSS software was used to perform these analyses, and the results were considered statistically significant when *p* value < 0.05. 

## 3. Results

The content (low, medium or high) of total fat, saturated fat, sugars and salt adjusted per 100 g of food or 100 mL of beverage was estimated, according to the FoP guide (Figure 1). In food products, high or moderate content of these nutrients was observed as predominant. Fat and saturated fat especially emerge as excessive. In contrast, very few beverage items high to moderate in these nutrients were observed, except for sugar.

More than 60% of the products offered in the VMs are beverages. The most frequent food categories found in the analyzed machines were “chocolates and bars” (10.6%) and “breadsticks” (8.4%), while the most frequent drink categories were “water” (27.7%) and “soft drinks” (23.4%). Products such as fruits, dried fruits, vegetable juices, milk and yogurts were absent. There were some differences in category distribution depending on the machine location, but they were only statistically significant for energy drinks and soft drinks (*p* < 0.05) (Table 2).

In addition, mean scores and standard deviation were calculated for each of the items reported in the nutritional label of the selected products in each category, per 100 g or 100 mL and per serving, expressed in grams, and their energy value, expressed in kcal (Table 3). Values per 100 g or 100 mL were also compared with those regarding the product serving size available in the VMs. Finally, the presence of palm oil-free and gluten-free products was assessed. Two out of three food products are free of palm oil. Products containing this oil are mainly in the categories of “pastries” and “chocolates and bars”. One out of three is a gluten-free food product, mainly belonging to the categories of “nuts”, “potato chips” and “candies and sweets”. All beverages were palm oil-free and gluten-free.

## 4. Discussion

Beverages were healthier than food products. Almost two-thirds of beverages could be considered healthy since the four nutrients associated with increased risk of chronic disease are low. One-third corresponds to unhealthy beverages, which are mainly sugar-sweetened soft drinks, energy drinks and milkshakes. Although beverages represent almost 60% of the products in VMs (Table 2), there is a wider range of food products than beverages. Only water and soft drinks account for 80% of beverage items. This aligns with consumer statistics which state that most purchased products in VMs were cold beverages, water being the leading product [9]. All VMs on the campus offer water, except for the Faculty of Philology, whose main beverage categories available were milkshakes and energy drinks, increasing the VMs’ total content of saturated fats and added sugars, respectively. Similar results have been observed in other studies [4,6,7,8,11,12,13].

There were some differences in category distribution depending on the machine location, but they were only statistically significant for energy drinks and soft drinks (*p* < 0.05) (Table 2). These differences can be attributed to the student’s demands and preferences. A review [14] reflected that associations between higher nutrition knowledge and dietary choices are significantly positive but weak.

A systematic review has been reported. Nutrition knowledge is a key component of health literacy [15]; therefore, health science faculties would be expected to contain a greater number of healthier categories than the humanities faculties. However, we found no evidence supporting this claim, which might be attributed to the fact that the faculties do not directly decide the products of their specific VMs, but rather the vending machine company with the people in the university in charge of this area choose the products that will be available for the whole university settings. 

The type of products available should be chosen to take their inclinations into account since they are the target population. Previous studies showed that both university staff and students desire healthier vending options and expressed the need to improve the nutritional quality of the food items sold in university VMs [16,17,18,19]. Most participants in these studies reported that they would purchase healthier products if available [16,17,18,19]. 

Palm oil is widely used in the food industry for its chemical/physical properties, low cost and wide availability. Its widespread use has led to an intense debate about whether it is a potential hazard to human health [20]. However, given the abundance of palm oil in the market, quantifying its true association with cardiovascular diseases is challenging [21]. Therefore, the matter is not palm oil itself but the fatty-acid-rich food group to which it belongs [20]. Two out of three food products available in the VMs assessed are free of palm oil, showing a greater concern of companies and the population about the use and consumption of this type of vegetable fats. Nevertheless, it should be mentioned that the most important thing is to consume less than 7–8% of saturated fatty acids, regardless of their origin, so an overall healthy diet should be prioritized for good cardiometabolic health [20].

One out of three foodstuffs in our VMs are gluten-free, mainly belonging to the categories of nuts, potato chips and candies and sweets, which are gluten-free by nature. Gluten-free products were absent in categories such as breadsticks, biscuits, sandwiches and bars. Patients with celiac disease account for 1% of the population, and its incidence is increasing [22]. The main treatment of celiac disease is based on a gluten-free diet, with an improvement of symptoms within days or weeks, generally. Despite its effectiveness, following a gluten-free diet has numerous difficulties due to the variable quality of information regarding the gluten-free status of food ingredients [22] and their availability. Thus, traditionally wheat-based categories should integrate new gluten-free alternatives in VMs to entice and retain clients with celiac disease. Several gluten-free foodstuffs contain more fat, including saturated fat, and salt than their equivalents with gluten [23]. However, to improve their nutritional quality, wholemeal gluten-free cereals and pseudocereals with high nutritive value should replace the low-nutritional gluten-free flours [23]. 

From the data gathered in Table 3 and considering the set of categories available in the VMs studied, the healthiest food categories in terms of nutrient content are sandwiches (with the lowest caloric value per 100 g and a moderate content of fats, including saturated fats, and a short number of added sugars) and nuts (with monounsaturated and polyunsaturated fats but with a high salt content and fried form). This last category could be improved by offering natural or toasted nuts without added salt, fats or sugars.

On the opposite side, the least suitable categories are chocolates and bars and pastries because of their high content of energy, fat, saturated fat and sugars. The high availability of these categories are the main contributors to nutritionally poor vending machines. It should be also considered that no fresh fruit was found in any of the analyzed VMs. 

In the case of beverages, the most appropriate category, and, therefore, the one that should predominate, is water, being the most purchased option by consumers as well [9]. Juices may also constitute a healthy option, provided that they are natural juices with no added sugars due to their vitamin content. The abuse of soft drinks with added sugars should be avoided, opting for those with 0% sugars, which are quite widespread among the different commercial brands increasing dental caries. If consumed excessively, they might also be detrimental to health [24].

It is convenient to consider the influence of the package or bottle size in the different categories offered in the VMs. This allows us to evaluate the amounts of nutrients being actually consumed by the clients since the purchase of these products usually entails their entire consumption on the same day. 

Most food categories have a serving size between 35 and 50 g, except for pastries (80 g) and sandwiches (140 g). The most energy-dense category is pastries due to their high amount of saturated fats and sugars. Pastries should be especially decreased in VMs due to their combination of large portions and unhealthy profiles. The second-most energy-dense category is sandwiches due to their large portion, but this category has a better nutritional profile since it provides less saturated fats and sugars. However, the nutritional profile should be improved to become a healthy choice since it still contains high amounts of salt. 

In the case of beverages, the bottles or cans volumes are between 200 mL and 500 mL. Thus, the amounts of nutrients are doubled and even quintupled. Consequently, vast amounts of sugar are being consumed with energy drinks, milkshakes and soft drinks. Therefore, reducing these categories will be key to improving the nutritional quality of VMs. 

Table 4 reflects our proposal to improve the nutritional quality of food and drinks available in VMs. While some studies have reported that most products sold in VMs are nutritionally poor [4,5,6,7,8], these have not offered specific recommendations that can be directly applied to VMs. Other studies have provided nutritional standards to guide the selection of VM products [25,26,27,28,29].

However, no study has provided key recommendations involving categories of products and their nutritional characteristics, which are informed based on the assessment of the nutritional quality. The application of standards could increase sales of healthy vending items and reduce sales of less healthy products [30]. While it is important that recommendations and standards are followed to improve the nutritional quality of VMs, it is also crucial to remember that VMs involve a business. Therefore, a balance between nutritional quality and profitability needs to be found. More strategies modifying availability, promotion and pricing should be explored further to maximize health benefits while minimizing revenue impacts [31]. It would be interesting to determine the impact of such interventions on overall sales. A previous study carried out by our group in 2000 at the same University reflected that the consumption of snacks was 27% of total dairy energy among university students [32]. In the present manuscript, the assessment of the nutritional composition of foods and beverages from vending machines indicates that VMs could contribute to an obesogenic environment and do not support healthy snacking. Although small differences exist between health science faculties and humanities faculties in terms of food categories, these could be explained by the convenience demands of students. Thus, no relationship between health knowledge and food choices was observed. 

On the other hand, the use of suggestions could help to guarantee the quality of VMs, including (i) recommendations to reduce proportions of nutrients associated with increased risk of chronic disease (NAIRCD) and (ii) recommendations to improve the nutritional profile of categories (Table 4). It will help to guarantee the improvement of the nutritional quality of vending machines. Furthermore, in our country, in 2008, the Spanish Network of Healthy Universities (REUS) [33] was created as an initiative that is part of an international strategy sponsored by the World Health Organization, which includes the CRUE (Conference of Rectors of Spanish Universities), the Ministry of Health, Social Services and Equality and the Ministry of Education, Culture and Sports. This initiative includes the ‘Specifications of technical specifications for vending machines and restaurant services’ and the ‘Declaration of universities as centers promoting health’. Currently, 55 Spanish universities are part of the network, including the University of Valencia. The use of recommendations reflected in Table 4 can help to guarantee this health initiative.

## 5. Conclusions

VMs should promote food categories such as fruits; natural or roasted nuts without added salt; whole-grain sandwiches, breadsticks and cereals that are low in sugar and salt. Foods made with olive oil instead of palm oil should be selected, and gluten-free alternatives should be introduced. VMs should promote beverage categories such as water, natural fruit and vegetable juices, milk, natural yogurts and sugar-free soft drinks while reducing sugar-sweetened beverages. These recommendations should be considered to develop official guidelines to improve the nutritional quality of VMs in university settings. Intervention studies should be performed to assess the implications and feasibility of these guidelines.

## Figures and Tables

**Figure 1 nutrients-14-05010-f001:**
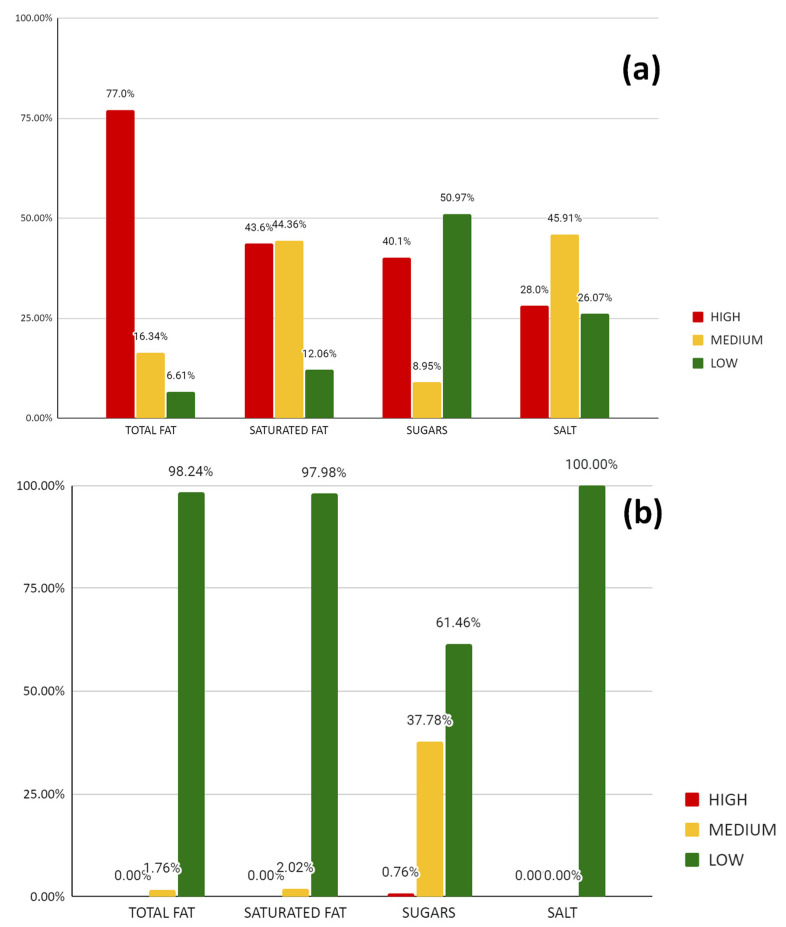
Percentage of foods with high, medium and low content of fats, saturated fat, sugars and salt expressed per 100 g of food product (**a**) and percentage of beverages with high, medium and low content of fats, saturated fat, sugars and salt expressed per 100 mL (**b**) in the whole campus. Classification according to the criteria in the Guide to creating a front-of-pack (FoP) nutritional label for pre-packed products sold through retail outlets.

**Table 1 nutrients-14-05010-t001:** Criteria for 100 g of food (whether or not it is sold by volume) and for drinks (per 100 mL) adapted from [10].

Text	Low	Medium	High	
Colour Code	Green	Amber	Red	
For food				
			>25% of RIs	>30% of RIs
Fat	≤3.0 g/100 g	>3.0 g to ≤17.5 g/100 g	>17.5 g/100 g	>21 g/portion ^1^
Saturates	≤1.5 g/100 g	>1.5 g to ≤5.0 g/100 g	>5.0 g/100 g	>6.0 g/portion ^1^
(Total) Sugars	≤5.0 g/100 g	>5.0 g to ≤22.5 g/100 g	>22.5 g/100 g	>27 g/portion ^1^
Salt	≤0.3 g/100 g	>0.3 g to ≤1.5 g/100 g	>1.5 g/100 g	>1.8 g/portion ^1^
For drinks				
			>12.5% of RIs	>15% of RIs
Fat	≤1.5 g/100 mL	>1.5 g to ≤8.75 g/100 mL	>8.75 g/100 mL	>10.5 g/portion ^2^
Saturates	≤0.75 g/100 mL	>0.75 g to ≤2.5 g/100 mL	>2.5 g/100 mL	>3 g/portion ^2^
(Total) Sugars	≤2.5 g/100 mL	>2.5 g to ≤11.25 g/100 mL	>11.25 g/100 mL	>13.5 g/portion ^2^
Salt	≤0.3 g/100 mL	>0.3 g to ≤0.75 g/100 mL	>0.75 g/100 mL	>0.9 g/portion ^2^

RIs: Reference Intakes. ^1^ Portion size criteria apply to portions/serving sizes greater than 100 g. ^2^ Portion size criteria apply to portions/serving sizes greater than 150 mL.

**Table 2 nutrients-14-05010-t002:** Percentage of each product category (food and drinks) compared to the total number of products in each faculty and the Campus as a whole.

	Humanities	Health Sciences	Campus Total	*p*-Value
Biscuits	4.3%	2.4%	3,1%	0.158
Breadsticks	8.3%	8.5%	8.4%	0.919
Candies and sweets	1.7%	0.5%	0.9%	0.105
Chewing gums	2.2%	1.7%	1.8%	0.634
Chocolates and bars	12.6%	9.4%	10.6%	0.207
Energy drinks	4.3%	0.9%	2.1%	0.004 *
Fruit	0.0%	0.0%	0.0%	1
Juices	3.9%	5.0%	4.6%	0.544
Milkshakes	3.0%	2.8%	2.9%	0.877
Nuts	3.0%	2.1%	2.4%	0.467
Pastries	2.6%	3.5%	3.2%	0.520
Potato chips	2.2%	1.9%	2.0%	0.802
Sandwich	0.0%	1.4%	0.9%	0.070
Snacks	5.7%	6.1%	6.0%	0.805
Soft drinks	16.1%	27.4%	23.4%	0.001 *
Water	30.0%	26.4%	27.7%	0.328

* Significantly different (*p* < 0.005).

**Table 3 nutrients-14-05010-t003:** Comparison of the nutritional markers of the different categories of food and beverages per 100 g or 100 mL, respectively.

	Energy (Kcal)	Fats (g/100 g/mL)	SFA (g/100 g/mL)	CH (g/100 g/mL)	Sugars (g/100 g/mL)	Fibra (g/100 g/mL)	Protein (g/100 g/mL)	Salt (g/100 g/mL)
Biscuits	471.7 ± 45.4	22.0 ± 4.6	8.5 ± 6.6	62.5 ± 3.2	23.2 ± 19.7	5.3 ± 3.3	7.7 ± 1.4	1.4 ± 1.6
Breadsticks	500.9 ± 17.4	25.5 ± 4.3	3.5 ± 2.1	56.6 ± 5.7	2.5 ± 2.8	3.6 ± 1.1	9.8 ± 1.0	1.8 ± 0.5
Candies and sweets	368.0 ± 24.0	1.6 ± 1.4	1.0 ± 0.9	86.7 ± 6.4	56.7 ± 20.7	0.0 ± 0.0	2.6 ± 2.5	0.1 ± 0.1
Chewing gum	155.0 ± 9.1	0.1 ± 0.2	0.0 ± 0.0	64.1 ± 2.8	0.0 ± 0.0	0.0 ± 0.0	0.0 ± 0.0	0.0 ± 0.0
Chocolates and bars	515.2 ± 50.6	28.1 ± 9.5	16.3 ± 5.9	57.5 ± 10.9	42.8 ± 9.9	2.7 ± 3.3	7.0 ± 1.3	0.7 ± 1.3
Nuts	566.2 ± 59.1	43.1 ± 12.3	5.3 ± 2.1	21.9 ± 19.9	7.4 ± 6.8	7.0 ± 3.6	20.1 ± 6.8	0.9 ± 0.8
Pastries	454.5 ± 45.2	23.8 ± 5.2	14.8 ± 4.4	54.7 ± 3.1	30.6 ± 6.6	2.9 ± 0.5	4.0 ± 1.2	0.5 ± 0.1
Potato chips	540.8 ± 16.2	34.2 ± 3.3	3.7 ± 0.6	49.9 ± 4.4	0.7 ± 0.2	3.1 ± 1.1	6.6 ± 1.2	1.1 ± 0.6
Sandwich	256.0 ± 18.7	12.5 ± 1.1	3.2 ± 1.4	26.5 ± 3.4	3.1 ± 0.9	0.0 ± 0.0	8.6 ± 1.7	1.6 ± 0.3
Snacks	484.7 ± 29.2	21.7 ± 4.5	2.5 ± 0.8	64.5 ± 3.9	3.2 ± 2.1	3.1 ± 1.4	6.0 ± 0.9	1.7 ± 0.6
Energy drinks	42.0 ± 13.8	0.0 ± 0.0	0.0 ± 0.0	10.1 ± 3.3	9.8 ± 3.5	0.0 ± 0.0	0.3 ± 0.2	0.2 ± 0.0
Juices	21.3 ± 15.1	0.0 ± 0.0	0.0 ± 0.0	4.6 ± 3.7	4.2 ± 3.2	0.4 ± 0.3	0.4 ± 0.1	0.0 ± 0.0
Milkshakes	62.8 ± 15.6	1.4 ± 0.9	0.9 ± 0.6	8.8 ± 3.0	8.5 ± 2.9	0.6 ± 0.5	3.5 ± 0.4	0.2 ± 0.0
Soft drinks	19.1 ± 14.6	0.0 ± 0.0	0.0 ± 0.0	4.6 ± 3.7	4.6 ± 3.7	0.0 ± 0.0	0.0 ± 0.1	0.0 ± 0.0
Water	0.0 ± 0.0	0.0 ± 0.0	0.0 ± 0.0	0.0 ± 0.0	0.0 ± 0.0	0.0 ± 0.0	0.0 ± 0.0	0.00 ± 0.0

**Table 4 nutrients-14-05010-t004:** Recommendations for the improvement of the nutritional quality of vending machines.

Key Recommendations to Reduce Risk of Chronic Disease (NAIRCD)	
Fat and saturated fat	Reduce food categories high in fat and saturated fat such as chocolates, pastries, snacks and potato chips. Consider reducing breadsticks and biscuits if their profile is not improved.
	Promote food groups that are low in fat and especially low in saturated fat such as fruits, salads and sandwiches. Consider the promotion of natural nuts as a source of healthy fats.
Sugars	Reduce sugar-rich food items such as chocolates, pastries and sweets.
	Promote food categories with moderate to low sugar content such as nuts, sandwiches and breadsticks if their profile is improved.
	Promote water and beverages with “low sugar” and “0% added sugar” claims.
Salt	Reduce categories with high or moderate salt that would be mainly: breadsticks, biscuits, snacks and sandwiches, given their profile was not improved.
	Promotion of the food categories with low salt content: nuts (without added salt) and fresh fruit.
Food items	Reduce fried salty nuts and introduce natural or roasted nuts without added salt. Some nuts with this profile are already present in the VMs.
	Substitute chocolates and bars for whole grain bars with seeds or dried fruits, preferably low low-moderate in saturated fat.
	Reduce potato chips preferably. Preference for chips fried in olive oil since it contains healthy monounsaturated fats.
	Promote snacks moderate in fat (mainly with monounsaturated and polyunsaturated fats), moderate–low in saturated fat, low in sugar and preferably moderate–low in salt. No snacks were currently offered with this nutritional profile.
	Introduce fresh fruits: they have high nutritional value due to their contribution of vitamins and minerals. Some high in fiber also provide satiety, which makes them an ideal snack.
	Although biscuits are not advisable, it is preferable to select wholegrain biscuits with moderate-low content of saturated fat, sugar and salt.
	Introduce wholegrain sandwiches and breadsticks moderate-low in salt, preferably with high proportions of nuts and seeds.
Beverages	Promote water over all other beverages. At least 60% of the drinks should be water.
	Reduce soft drinks. Choose sugar-free or low in sugar options.
	Promote milkshakes and juices with a higher percentage of milk, fruits and vegetables in their ingredients.
	Reduce energy drinks and choose those low in sugar.
	Introduce categories such as milk or yogurts with fruits and without added sugars.

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
