# Peer review of "Assessment of Nutritional Quality of Products Sold in University Vending Machines According to the Front-of-Pack (FoP) Guide"

_nutrients, 2022, doi:10.3390/nu14235010_

Round 1

Reviewer 1 Report

It would have been interesting to know the amount and type of products consumed  in 24 hours. 

Author Response

Comments and Suggestions for Authors:  It would have been interesting to know the amount and type of products consumed  in 24 hours.

Author’s comment: Thank you for your comment and we think that it is a great idea. The amount and type of products consumed in 24 hours are not the objective of this study. The study addresses the nutritional composition of food and beverages available through vending machines (VMs) in a university campus, measuring the adequacy with current recommendations. However, a study developed by our group in 2000 reflected that 27% of total dairy energy at the consumption was of snacks among university students.  We have added this information in the discussion section.

Reviewer 2 Report

Suggestion: the methodology should include the cutoffs used for high, medium and low for saturated fats, salt and sugars and not just refer to the Classification according to the criteria in the Guide to creating a front of pack (FoP).

Author Response

Suggestion: The methodology should include the cutoffs used for high, medium and low for saturated fats, salt and sugars and not just refer to the Classification according to the criteria in the Guide to creating a front of pack (FoP).

Author’s comment: According to your comment, we have added this idea in the method section.

Reviewer 3 Report

The study addresses the nutritional composition of food and beverages available through vending machines (VMs) in a university campus, measuring the adequacy with current recommendations. Considering than VMs are expected to contribute to obesity, there is need for better selection of items available to students and workers inside campus, not different between various scientific areas. I have a few questions that merit discussion:

1.     The title is definitely misleading. There is no evidence that this is the case, on the basis of the present report. The report is merely an observation of what is offered, without any direct implication for obesity epidemics. This does not mean that the problem should be overcome, but should be presented in the discussion as a reasonable hypothesis.

2.     The study provides a reasonable analysis of the type of food items offered in the VMs, but there is no mention of the quantity of individual item selection. It would be very important, if the authors would engage in a systematic analysis of the importance of VMs in the obesogenic environment, to know customer preferences. Obesity is the product of obesogenic content times frequency of use.

3.     It seems that VMs are specifically addressing the preferences of customers. Frankly I do not believe that students of history and geography need a larger energy supply to cope with longer study hours (lines 168-69). Similarly, the use of energy drinks in other areas is more an effect of advertisement than a real need (line 169-70).

4.     Line 78: MVs instead of VMs.

5.     The discussion is largely repetitive and might be cut by half without any significant loss of information. It is scarcely put in the context of the present literature.

6.     The manuscript would definitely improve by a larger discussion of the importance of VMs in addressing the need of a less obesogenic nutrition. VMs might accomplish the needs for nudging towards healthier food products, as expressed in the final Table. In order to do that, we need restrictive regulation for the provision of foods inside VMs, which also collide with market economy.

Author Response

Comments and Suggestions for Authors:  The study addresses the nutritional composition of food and beverages available through vending machines (VMs) in a university campus, measuring the adequacy with current recommendations. Considering than VMs are expected to contribute to obesity, there is need for better selection of items available to students and workers inside campus, not different between various scientific areas. I have a few questions that merit discussion:     The title is definitely misleading. There is no evidence that this is the case, on the basis of the present report. The report is merely an observation of what is offered, without any direct implication for obesity epidemics. This does not mean that the problem should be overcome, but should be presented in the discussion as a reasonable hypothesis.

Author’s comment: According to your comment, the title has been changed from ‘Do University Vending Machines Contribute to Obesogenic Environment?’ to ‘Assessment of nutritional quality of products sold in university vending machines according to the Front of Pack (FoP) Guide’. Furthermore, the previous title is presented in the discussion as reasonable hypothesis.

Comments and Suggestions for Authors:  The study provides a reasonable analysis of the type of food items offered in the VMs, but there is no mention of the quantity of individual item selection. It would be very important, if the authors would engage in a systematic analysis of the importance of VMs in the obesogenic environment, to know customer preferences. Obesity is the product of obesogenic content times frequency of use.

Author’s comment: This a good idea, but it was not aim of the manuscript. We can keep it in mind for the future.

Comments and Suggestions for Authors:  It seems that VMs are specifically addressing the preferences of customers. Frankly I do not believe that students of history and geography need a larger energy supply to cope with longer study hours (lines 168-69). Similarly, the use of energy drinks in other areas is more an effect of advertisement than a real need (line 169-70).

Author’s comment: According to your comment, we have deleted this idea to clarify it.

Comments and Suggestions for Authors:  Line 78: MVs instead of VMs.

Author’s comment: According to your comment, it has been changed.

Comments and Suggestions for Authors:  The discussion is largely repetitive and might be cut by half without any significant loss of information. It is scarcely put in the context of the present literature.

Author’s comment: It has been modified and cut.

Comments and Suggestions for Authors:  The manuscript would definitely improve by a larger discussion of the importance of VMs in addressing the need of a less obesogenic nutrition. VMs might accomplish the needs for nudging towards healthier food products, as expressed in the final Table. In order to do that, we need restrictive regulation for the provision of foods inside VMs, which also collide with market economy.

Author’s comment: According to your comment, we have added the Spanish food politics about VMs which used as internal regulation among Spanish Universities. It has added in the discussion section.
